# Functional and Morphological Outcomes after Trabeculectomy and Deep Sclerectomy—Results from a Monocentric Registry Study

**DOI:** 10.3390/diagnostics14010101

**Published:** 2024-01-02

**Authors:** Valentin Pfeiffer, Pascal Aurel Gubser, Xiao Shang, Joel-Benjamin Lincke, Nathanael Urs Häner, Martin Sebastian Zinkernagel, Jan Darius Unterlauft

**Affiliations:** University Eye Hospital, Inselspital, University of Bern, Freiburgstrasse 18, 3010 Bern, Switzerland; valentin.pfeiffer@insel.ch (V.P.); pascal.gubser@students.unibe.ch (P.A.G.); xiao.shang@students.unibe.ch (X.S.); joel-benjamin.lincke@insel.ch (J.-B.L.); nathanael.haener@insel.ch (N.U.H.); martin.zinkernagel@insel.ch (M.S.Z.)

**Keywords:** glaucoma, glaucoma surgery, trabeculectomy, deep sclerectomy, optical coherence tomography (OCT), visual field, perimetry

## Abstract

The aim of this study was to compare the effectiveness of trabeculectomy (TE) and deep sclerectomy (DS) in lowering intraocular pressure (IOP) and thereby preserving visual field and peripapillary retinal nerve fiber layer (RNFL) tissue in primary open-angle glaucoma (POAG) cases. IOP, number of IOP-lowering medications, visual acuity, mean defect of standard automated perimetry, and mean peripapillary RNFL thickness were retrospectively collected and followed up for 3 years after surgery. TE was performed in 104 eyes and DS in 183 eyes. Age, gender, laterality, IOP, number of medications, visual acuity, perimetry mean defect, and peripapillary RNFL thickness were equally distributed at baseline. Mean IOP decreased from 23.8 ± 1.4 mmHg and 23.1 ± 0.4 mmHg to 13.4 ± 0.6 mmHg (*p* < 0.001) and 15.4 ± 0.7 mmHg (*p* = 0.001) in the TE and DS groups, respectively. Mean defect remained stable (TE: −11.5 ± 0.9 dB to −12.0 ± 1.1 (*p* = 0.090); DS: −10.5 ± 0.9 dB to −11.0 ± 1.0 dB (*p* = 0.302)), while mean peripapillary RNFL thickness showed further deterioration during follow-up (TE group: 64.4 ± 2.1 μm to 59.7 ± 3.5 μm (*p* < 0.001); DS group: 64.9 ± 1.9 μm to 58.4 ± 2.1 μm (*p* < 0.001)). Both TE and DS were comparably effective concerning postoperative reduction in IOP and medication. However, glaucoma disease further progressed during follow-up.

## 1. Introduction

Glaucoma is a heterogeneous group of eye diseases in which the progressive demise of retinal ganglion cells (RGCs) leads to the development of visual field defects and ultimately leads to (unilateral or bilateral) blindness if left untreated or insufficiently treated for too long [1]. The only known treatment option with proven effectiveness in slowing down the natural disease process is a reduction in intraocular pressure (IOP) [2,3,4,5]. IOP can be lowered by means of medical treatment or surgery. Surgical intervention is oftentimes deferred for cases unresponsive to medication, cases with intolerance to medication, and cases with very high/decompensated IOPs.

Trabeculectomy in its current form was first described by Sugar and Cairns in the 1960s and has since been considered the gold standard for surgical glaucoma treatment [6]. This is mostly due to the proven effectiveness in reducing even the highest IOP to physiological regions. Apart from this, TE also has a number of limitations and possible occurring complications, such as early- and late-onset bleb-related problems, which potentially lead to vision loss [7,8,9]. One major drawback of TE is the potential for postoperative hyperfiltration with possible long-lasting hypotonia with induction of maculopathy and vision loss. Deep sclerectomy (DS) is another surgical technique to reduce IOP. DS is very similar to TE but functions through the percolation of aqueous humor through the trabecular meshwork, which, although deroofed during surgery, is mostly left intact [10]. Aqueous humor percolates into a preformed intrascleral reservoir and flows from here beneath a scleral flap into the subconjunctival/subtenonal space. By this mode of action, hypotonia-related complications are, in theory, avoided. Rulli et al. performed a systematic review of a large number of already published studies and compared the reported rates for the most frequently occurring complications after TE and DS [7]. They found that the probability for hypotony, choroidal effusion, flat anterior chamber, and cataract development is higher after TE than DS.

It was the aim of our study to compare the postoperative results concerning IOP, medication, and visual acuity between TE and DS in a comparable group of primary open-angle glaucoma patients over a median length follow-up of 3 years. One central question was whether the IOP-lowering effect of DS is comparable to that observed after TE and whether this effect is comparatively long-lasting. Apart from this, it was a main aim to take functional and anatomical parameters (i.e., the visual field and mean peripapillary retinal nerve fiber layer thickness) into account when comparing both surgical techniques. To date, limited data exist concerning the long-term structural and functional outcomes after TE and DS. However, it has been shown before that after surgical glaucoma intervention, further RGC loss develops although sufficient IOP decrease [11].

## 2. Materials and Methods

The inclusion criteria of this study were as follows. The indication for surgical glaucoma intervention was usually given due to disease progression under full bearable medical treatment when IOP was fully decompensated and unresponsive to medical treatment or when (further) medical treatment was not tolerated due to known allergies or present contraindications against medical treatment. Since this was a retrospective study, the indication to perform TE or DS was highly reliant on surgeons’ preferences and did not follow strict decision paths. Furthermore, only cases of primary open-angle glaucoma were included, and a full medical history spanning the first three years after surgery had to be present. All surgical procedures were performed by four experienced ophthalmic surgeons. Both TE and DS were regularly performed by each of the four surgeons.

The utilized surgical techniques for TE and DS followed already published standards [12,13]. Both TE and DS were performed with a fornix-based conjunctival flap. Mitomycin C (0.2 mg/mL) was applied for 2 min and was then meticulously rinsed with 20 mL BSS. In TE cases, the scleral flap was 4 × 4 mm; in DS, it was 5 × 5 mm in size. In TE cases, the anterior chamber was entered using a diamond knife, and an iridectomy was performed using Vannas micro scissors. In DS, a second deep scleral flap of 2 × 2 mm was fashioned. After excision of the second scleral flap, Schlemm’s canal was delineated and deroofed. Following these steps, filtration of aqueous humor through the remaining trabecular meshwork was usually visible. As a last step, the inner portion of the trabecular meshwork was stripped. After these steps, the scleral flap was reapproximated using 2 to 5 nonabsorbable 10/0 single-button sutures. Number and tension of flap sutures were adapted to intraoperative visibility of protrusion of aqueous humor underneath the scleral flap and resulting IOP. Finally, Tenon and conjunctiva were reapproximated to the limbus using 2 to 4 absorbable single-button sutures and 1–2 nonabsorbable mattress sutures to assure water tightness.

During postoperative course, laser suture lysis or subconjunctival injection of 5-fluorouracil (5-FU; 0.1 mL with a concentration of 50 mg/mL) or needling procedures were performed as needed depending on bleb morphology on slit lamp examination or course of IOP. All these were usually only performed during the first 3 to 6 months after surgery in both groups. The 5-FU injection was performed with the patient sitting in a heads-up gaze downward position. A needling procedure was usually performed under observation with an operating microscope with the patient in a supine position.

Medical treatment regimens after surgery were similar in both TE and DS groups. Medical treatment started with both topical antibiotics (tobramycin, QID for 3 weeks), steroids (prednisolone acetate, QID for 3 weeks), and cycloplegics (atropine 0.5%, BID for 1 week). Usually, at the follow-up visit 3 weeks after surgery, the nonabsorbable mattress suture was taken out, after which antibiotic treatment was also stopped. Topical steroid treatment was further continued and tapered depending on clinical assessment of postoperative inflammation and conjunctival scarification reaction. Follow-up examinations were usually scheduled weekly during the first month after surgery to monitor postoperative bleb development, search for signs of scar formation, and rapidly intervene when needed. During the further course, examinations were scheduled for 3, 6, 12, 24, and 36 months after surgery.

Demographic data such as gender, age, and laterality were collected before surgery. Data collected from baseline and follow-up examinations (usually scheduled 1 day as well as 1, 3, 6, 12, and 24 months after surgery) included the following: IOP (measured using Goldmann applanation tonometry, Haag-Streit, Köniz, Switzerland), number of applied IOP-lowering active medical compounds, BCVA (measured using Snellen charts and transformed to logMAR units for statistical analysis). Furthermore, mean defect of visual field testing (Octopus, Haag-Streit, Köniz, Switzerland), which was usually performed at the follow-up visits at 6, 12, 24, and 36 months, as well as mean peripapillary RNFL thickness as measured using SD-OCT (Spectralis OCT, Heidelberg Engineering, Heidelberg, Germany), which was usually performed at the 3-, 6-, 12-, 24-, and 36-month postsurgical visits, were collected. Visual field test results were only used for analysis when reliability criteria were met. These were a false-positive and false-negative rate of <20%. Success was defined as a minimal IOP reduction of 20% compared with baseline with (qualified success) or without (complete success) application of IOP-lowering medication and a resulting IOP of <21 mmHg, <18 mmHg, or <15 mmHg.

Data capturing, editing, and analysis were performed using Excel 2016, Version 16.0, release date 2015 (Microsoft, Redmond, Washington, DC, USA) and SPSS programs (IBM, Version 24.0; Chicago, IL, USA). For description, continuous variables are given as mean and standard error of mean. Categorical variables are furthermore described as frequencies. For statistical analysis, chi-square-tests and nonparametric Wilcoxon and Mann–Whitney *U*-tests were performed. A *p*-value of less than 0.05 was considered statistically significant.

## 3. Results

The medical files of all eyes undergoing TE or DS between 2014 and 2019 at the University Eye Hospital Bern, Inselspital, Switzerland, were reviewed and included for analysis when the aforementioned inclusion criteria were met. For analysis, 104 cases of TE and 183 cases of DS were included. The mean patient age was 63.5 ± 1.6 years in the TE group and 64.9 ± 1.1 years in the DS group (*p* = 0.710). A total of 53 patients in the TE group and 101 patients in the DS group (*p* = 0.285) were female (Table 1). In the TE group, the right eye underwent surgery in 46 cases, and in the DS group, the right eye underwent surgery in 95 cases (*p* = 0.126).

The mean presurgical IOP was 23.8 ± 1.4 mmHg (10–55 mmHg) in the TE group and 23.1 ± 0.4 mmHg (6–58 mmHg) in the DS group (*p* = 0.889). In the TE group, the mean IOP decreased to 10.4 ± 0.6 mmHg (*p* < 0.001) at the 1-month follow-up after surgery (Figure 1 and Table 2). Thereafter, the mean IOP slowly increased during follow-up to 11.9 ± 1.0 mmHg (*p* < 0.001), 12.9 ± 1.2 mmHg (*p* < 0.001), and 13.4 ± 1.4 mmHg (*p* < 0.001) at 1, 2, and 3 years after surgery. In the DS group, the mean IOP decreased to 10.8 ± 0.3 mmHg (*p* < 0.001) at the follow-up examination 1 month after surgery. During the further postoperative course, the mean IOP slightly increased to 14.4 ± 0.5 mmHg (*p* < 0.001), 14.6 ± 0.5 mmHg (*p* < 0.001), and 15.4 ± 0.7 mmHg (*p* < 0.001) at the follow-up examinations 1, 2, and 3 years after surgery. Comparison of the mean IOPs between the TE and DS groups demonstrated a difference of statistical significance for results measured at the follow-up examination 1 (*p* = 0.009) and 3 years (*p* = 0.037) after surgery (Table 2).

The mean number of IOP-lowering medications followed a comparable course as IOP during follow-up (Figure 2 and Table 2). The mean number of medications at baseline was 3.2 ± 0.2 in the TE group and 3.3 ± 0.1 in the DS group (*p* = 0.182). In the TE group, the mean number of IOP-lowering medications dropped to 0.5 ± 0.1 (*p* < 0.001) at the follow-up examination 1 year after surgery and increased slightly thereafter to 0.8 ± 0.1 (*p* < 0.001) and 0.9 ± 0.2 (*p* < 0.001) at the 2- and 3-year follow-up visits, respectively. In the DS group, the mean number of taken IOP-lowering medications followed a course comparable to that in the TE group. The mean number of medications was 0.7 ± 0.1 (*p* < 0.001), 1.1 ± 0.1 (*p* < 0.001), and 1.4 ± 0.1 (*p* < 0.001) at the visits 1, 2, and 3 years after surgery. When comparing the number of IOP-lowering medications between both groups, a difference of statistical significance was found for the follow-up examinations at 2 (*p* = 0.024) and 3 (*p* = 0.008) years after surgery.

Visual acuity stayed reasonably stable in the TE and DS groups during the first three postoperative years of follow-up (Figure 3 and Table 2). Preoperatively, the mean BCVA was of comparative strength in both groups (TE: 0.30 ± 0.02 logMAR; DS: 0.34 ± 0.02 logMAR; *p* = 0.302). In both groups, BCVA dropped strongly after surgery but reached preoperative values again at the follow-up examination 3 months after surgery (TE: 0.38 ± 0.03 logMAR; DS: 0.41 ± 0.03 logMAR; *p* = 0.895). From the follow-up examination 3 months after surgery onwards, BCVA remained stable, and neither differences in preoperative values nor differences between both groups showed statistical significance at any further follow-up examination. At 36 months after surgery, the mean BCVA was 0.37 ± 0.04 logMAR in the TE group and 0.41 ± 0.05 logMAR in the DS group (*p* = 0.234).

Mean deviation measured using standard automated perimetry (SAP) was comparable between groups at baseline (TE: −11.5 ± 0.9 dB; DS: −10.5 ± 0.9 dB; *p* = 0.377/for MD distribution before surgery; see Figure 4). During 3 years of follow-up, the mean MD further decreased mildly in the TE and the DS group but did not show differences of statistical significance when compared with baseline values (Figure 5 and Table 3). Furthermore, a comparison of the TE and DS groups did not reveal differences of statistical significance either at baseline or at the follow-up examinations (Table 3).

Mean peripapillary RNFL thickness, as assessed using circularly oriented SD-OCT scans at baseline, was 64.4 ± 2.1 μm in the TE group and 64.9 ± 1.9 μm in the DS group (*p* = 0.963). Mean RNFL thickness dropped to 59.7 ± 3.5 μm (*p* < 0.001; compared with baseline) in the TE group and 58.4 ± 2.1 μm (*p* < 0.001; compared with baseline) in the DS group at the follow-up examination 3 years after surgery. Comparison of the mean peripapillary RNFL thickness between the TE and DS groups at the follow-up examination 1 year after surgery did not reveal differences of statistical significance (*p* = 0.620). From the follow-up examination 12 months after surgery onwards, the mean RNFL thickness seemed to remain reasonably stable in the TE and DS groups (Figure 6 and Table 3). Comparison of the mean RNFL thickness, as measured at the follow-up examination at 24 months and 36 months, still showed differences of statistical significance when compared with baseline values in the TE and DS groups (for details, see Table 3). However, when compared with the respective preceding measurements at 24 and 36 months after surgery, no difference of statistical significance was detectable anymore (TE: 12 months ≈ 24 months: *p* = 0.131; 24 months ≈ 36 months: *p* = 0.842//DS: 12 months ≈ 24 months: *p* = 0.374; 24 months ≈ 36 months: *p* = 0.079).

A number of occurring complications and necessary postoperative interventions during follow-up accompanied both performed surgical techniques. These included minor complications like corneal erosions and hyphema formation immediately after surgery. Intermediary complications like bleb leakage and transient choroidal detachment were more frequent in the TE than in the DS group, dissolving in all cases without further necessary intervention (Table 4). Further vision-threatening complications like hypotonia maculopathy and aqueous misdirection syndrome were equally distributed in both groups. Suture lysis and 5-FU use were more pronounced in the TE group, and goniopuncture was naturally only performed in DS eyes. However, the necessity to perform a secondary surgical glaucoma intervention was equally distributed in both groups (*p* = 0.875). During the 3 years of follow-up in the TE group, three eyes underwent cyclophotocoagulation, seven eyes received a tube shunt implant (three Baerveldt, four Ahmed), and in four further eyes, TE was repeated. In the DS group, 16 eyes underwent TE, 3 eyes received a Preserflo microshunt, 2 eyes received a Baerveldt tube shunt, and in 2 eyes, a complete open revision of the DS was performed.

After 3 years of follow-up, 72%, 72%, and 70% of eyes in the TE group reached complete success with resulting IOP levels of <21 mmHg, <18 mmHg, and <15 mmHg, respectively. In the DS group, the percentages of eyes reaching complete success and a final resulting IOP of <21 mmHg, <18 mmHg, and <15 mmHg were 46%, 45%, and 40%. Further statistical analysis revealed that the different percentages of eyes reaching complete and qualified success levels in the TE and DS groups were of statistical significance for each IOP level analyzed (Table 5 and Figure 7).

## 4. Discussion

With the presented results, it was shown that using TE and DS IOP and the number of IOP-lowering medications can be lowered with a lasting effect in the medium term during the first 3 years after surgery. However effective, both IOP and the number of medications increased slowly during postsurgical follow-up after TE and DS. After 3 years of follow-up, the mean IOP and mean number of IOP-lowering medications were higher in the DS group than in the TE group. The percentage of eyes reaching both complete and qualified success was higher in the TE group after 3 years.

Apart from the results that describe nicely the aforementioned effectiveness of TE and DS concerning IOP development, we also demonstrated a further developing RNFL decrease after surgery in both groups. Another central finding was that the velocity of postsurgical RNFL decreased and slowed down during follow-up in both the TE and DS groups. While an RNFL thickness decrease developed seemingly unbroken during the first year after surgery, RNFL thickness stabilized during the second and third years after surgery. Interestingly, all these developments took place while the SAP and BCVA results remained stable.

The IOP-lowering effectivity of TE has been proven numerous times before in a number of single- and multicenter trials [14,15,16,17,18,19]. Maybe the two most prominent examples describing the effectiveness of TE are those published by Edmunds and colleagues and Kirwan and colleagues. Edmunds and colleagues [15], using a nationwide multicenter survey conducted in the UK in the 1990s, showed by analysis of clinical courses of >1200 eyes undergoing TE a mean IOP decrease from 26.2 to 14.4 mmHg, with 66.6% of eyes reaching an IOP decrease of 1/3 compared with presurgical baseline values without the need to apply additional IOP-lowering medications. Kirwan and colleagues [19] performed a similar nationwide multicenter survey in the UK in the early 2000s including the results of >400 eyes and demonstrated a mean IOP decrease from 23.0 ± 5.5 mmHg to 12.4 ± 4.0 mmHg and a decrease in the mean number of IOP-lowering medications from 2.5 ± 0.9 to 0.1 ± 0.4 during 24 months of postsurgical follow-up. These developments are comparable to the results observed in our TE group. However, we added further descriptions of functional (visual field test) and anatomical (RNFL thickness) development after surgery, adding to the already published knowledge about the effectiveness of TE [11,20,21].

DS was developed in an attempt to make surgical glaucoma interventions (i.e., TE and its precursors) safer by reducing the probability of the occurrence of hypotonia, choroidal detachment, and hypotonia-induced vision loss by leaving the trabecular meshwork mostly intact [22]. DS functions through the induction of a transtrabecular percolation of aqueous humor from the anterior chamber into a preformed intrascleral reservoir and Schlemm’s canal, thereby reducing IOP. To date, a number of studies exist that have directly compared the effectiveness of lowering IOP and the necessity for the application of IOP-lowering medications after both interventions [7,23,24]. Most of these comparative studies favored TE over DS in view of a reduction in IOP and IOP-lowering medications. A Cochrane analysis performed by Eldaly and colleagues showed that both methods were comparatively effective in lowering the IOP and medication burden. However, TE led to more postoperative complications than DS, which were mostly mild and responded well to treatment [25]. Our data showed that minor and intermediate complications were, although rare, more frequent in the TE group and that graver complications were equally distributed in the TE and DS groups. However, further necessary interventions like 5-FU applications and laser suture lysis were deemed necessary in the TE group more often, and it could be argued how large the impact of both might have been on the final IOP, number of medications and percentage of successful cases, which were all in favor of TE. Finally, most studies on DS demonstrated effectiveness in shorter or longer postoperative follow-up terms, with most not giving detailed functional (i.e., visual field) or anatomical results (i.e., RNFL scans) [23,26,27,28,29,30].

Some prospective randomized clinical trials have already been performed comparing the effectiveness of TE and DS [26,31,32]. Cillino and colleagues [31] compared the postoperative results of 21 eyes undergoing TE and 19 eyes scheduled for DS and found a comparable decrease in the mean IOP from 28.0 ± 6.0 to 16.1 ± 3.0 mmHg 1 year after surgery in the TE group and from 29.6 ± 5.8 to 14.5 ± 5.0 mmHg in the DS group. Neither before (*p* = 0.48) nor after 12 months (*p* = 0.53) was a difference of statistical significance demonstrated between the TE and DS groups concerning mean IOP. Furthermore, Russo and colleagues [32] demonstrated that although reporting comparable results concerning an IOP decrease 48 months after surgery, the levels of complete success were significantly higher in the TE than in the DS group (72% vs. 51%; *p* < 0.05). We are the first to describe differences of statistical significance for IOP and medication results between eyes that underwent TE or DS during 3 years of postsurgical follow-up. Concerning IOP, the differences were statistically significant at the follow-up examinations 1 and 3 years after surgery. Different medication use was statistically significant at 2 and 3 years after surgery. Finally, the percentage of eyes reaching qualified and complete success levels was higher in our followed TE group, and the difference between both groups was statistically significant at 1, 2, and 3 years independent of the underlying resulting IOP levels (<21 mmHg, <18 mmHg, <15 mmHg).

Examination of the macula and the optic nerve head using OCT techniques has found its way into clinical routine, especially for counseling glaucoma patients, for several years now [33]. During follow-up after surgical glaucoma intervention, OCT parameters allow for IOP and the number of IOP-lowering medication development for an objective evaluation of results. However, development after intervention and ensuing IOP reduction in glaucoma seems to follow specific paths similar to other diseases that lead to optic nerve head atrophy (i.e., optic neuritis). The OCT data hereby presented suggest a further progression of RNFL loss even though IOP was lowered effectively through both TE and DS for the first year after surgery, stabilizing thereafter.

The described study’s results are not without weaknesses. Here, we described the analysis of a nonrandomized, unblinded retrospective monocentric cohort study. Therefore, a bias cannot be ruled out because neither patients nor examining staff were blinded to the performed interventions. However, we reported the results of the biggest cohort study to date, comparing both surgical interventions directly, with additional descriptions of functional (i.e., visual field tests) and anatomical results (i.e., peripapillary RNFL thickness as measured using OCT) for a median follow-up of 3 years after surgery.

## 5. Conclusions

TE and DS are both highly effective in lowering IOP and medication burden in glaucoma for a medium-length term while stabilizing visual field parameters and RNFL demise. Both techniques are accompanied by a number of complications and frequently necessitate secondary interventions while being comparatively safe. However, when comparing both techniques, TE led to a lower IOP and medication burden, and a higher fraction of eyes reached success levels at 1, 2, and 3 years after surgery.

## Figures and Tables

**Figure 1 diagnostics-14-00101-f001:**
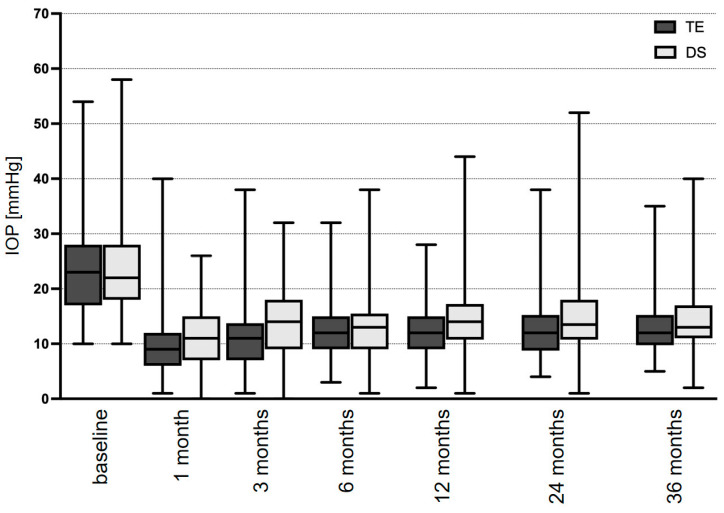
Mean IOP development over the course of 3 years after TE or DS: TE, trabeculectomy; DS, deep sclerectomy.

**Figure 2 diagnostics-14-00101-f002:**
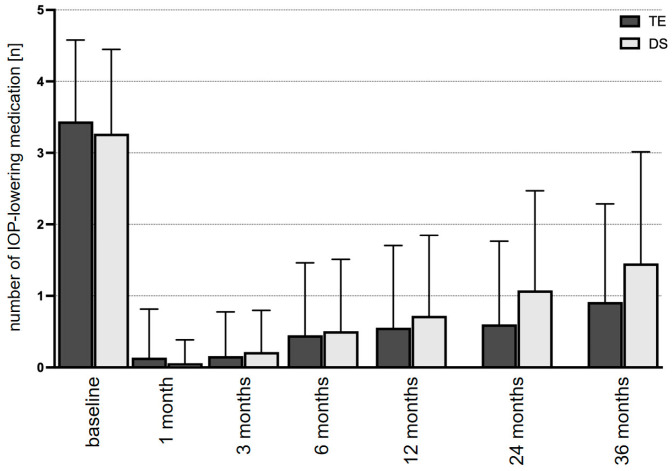
Development of the mean number of necessary IOP-lowering medications over the course of 3 years after TE or DS: TE, trabeculectomy; DS, deep sclerectomy.

**Figure 3 diagnostics-14-00101-f003:**
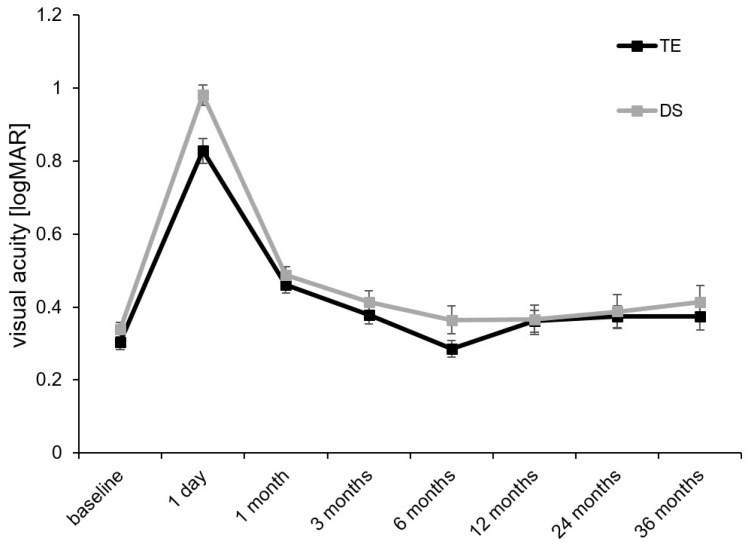
Course of the mean BCVA over the 3 years of follow-up after TE or DS: TE, trabeculectomy; DS, deep sclerectomy.

**Figure 4 diagnostics-14-00101-f004:**
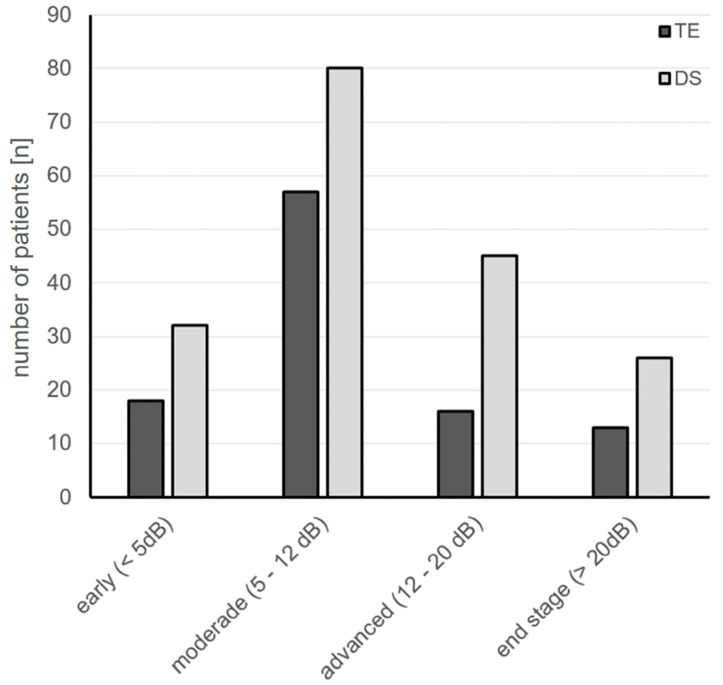
Distribution of SAP mean deviation in groups undergoing TE or DS at baseline before surgery: TE, trabeculectomy; DS, deep sclerectomy.

**Figure 5 diagnostics-14-00101-f005:**
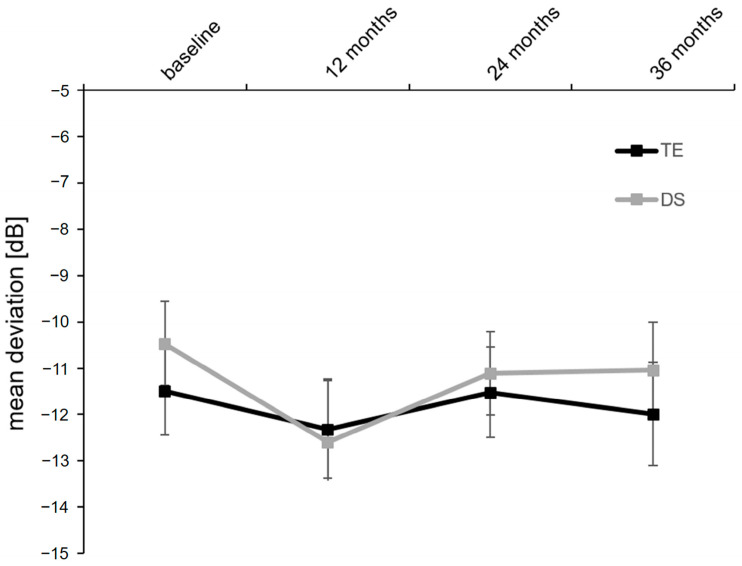
Course of the visual field mean deviation over the 3 years of follow-up after TE or DS: TE, trabeculectomy; DS, deep sclerectomy.

**Figure 6 diagnostics-14-00101-f006:**
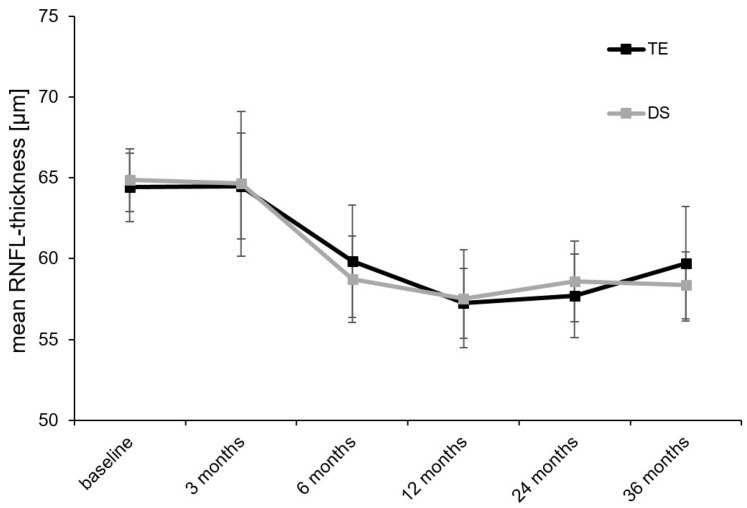
Course of the mean RNFL thickness measured using peripapillary OCT scan over the 3 years of follow-up after TE or DS: RNFL, retinal nerve fiber layer; OCT, optical coherence tomography; TE, trabeculectomy; DS, deep sclerectomy.

**Figure 7 diagnostics-14-00101-f007:**
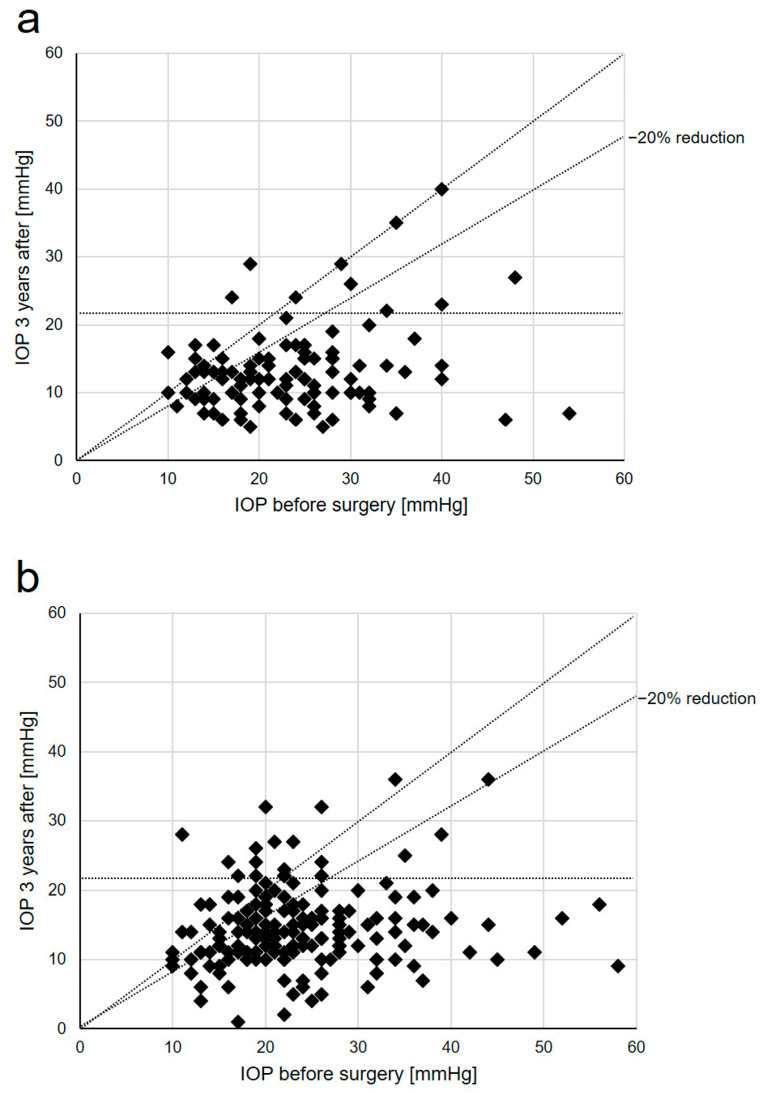
Scattergrams of pre- and postoperative IOP results in the TE (**a**) and DS (**b**) groups 3 years after the respective surgery: TE, trabeculectomy; DS, deep sclerectomy; IOP, intraocular pressure.

**Table 1 diagnostics-14-00101-t001:** Baseline characteristics of patients undergoing TE or DS.

	TE	DS	*p* = *
Number of eyes, *n*	104	183	n.a.
Age, years	63.5 ± 1.6	64.9 ± 1.1	0.710
Laterality, right/left	46/58	95/88	0.126
Gender, female/male	53/51	101/82	0.285
Baseline IOP, mmHg	23.8 ± 1.4	23.1 ± 0.4	0.889
Number of IOP-lowering medications, *n*	3.2 ± 0.2	3.3 ± 0.1	0.182
Visual acuity, logMAR	0.30 ± 0.02	0.34 ± 0.02	0.302
Visual field mean defect, dB	−11.5 ± 0.9	−10.5 ± 0.9	0.377
Mean peripapillary RFNL thickness, μm	64.4 ± 2.1	64.9 ± 1.9	0.963

TE, trabeculectomy; DS, deep sclerectomy; IOP, intraocular pressure; RNFL, retinal nerve fiber layer; * (chi-square test).

**Table 2 diagnostics-14-00101-t002:** Development of IOP, IOP-lowering medication, and BCVA during 24 months of postsurgical follow-up after TE or DS.

		TE	*p* = *	DS	*p* = *	*p* = **
IOP, mmHg	Baseline	23.8 ± 1.4	n.a.	23.1 ± 0.4	n.a.	0.889
	3 months	10.8 ± 0.4	<0.001	13.7 ± 0.5	<0.001	0.002
	6 months	11.9 ± 0.4	<0.001	13.2 ± 0.4	<0.001	0.363
	12 months	11.9 ± 1.0	<0.001	14.4 ± 0.5	<0.001	0.009
	24 months	12.9 ± 1.2	<0.001	14.6 ± 0.5	<0.001	0.122
	36 months	13.4 ± 1.4	<0.001	15.4 ± 0.7	<0.001	0.037
Medication, *n*	Baseline	3.2 ± 0.2	n.a.	3.3 ± 0.1	n.a.	0.182
	3 months	0.1 ± 0.1	<0.001	0.6 ± 0.1	<0.001	0.142
	6 months	0.4 ± 0.1	<0.001	0.5 ± 0.1	<0.001	0.424
	12 months	0.5 ± 0.1	<0.001	0.7 ± 0.1	<0.001	0.158
	24 months	0.8 ± 0.1	<0.001	1.1 ± 0.1	<0.001	0.024
	36 months	0.9 ± 0.2	<0.001	1.4 ± 0.1	<0.001	0.008
BCVA, logMAR	Baseline	0.30 ± 0.02	n.a.	0.34 ± 0.02	n.a.	0.302
	3 months	0.38 ± 0.03	0.611	0.41 ± 0.03	0.389	0.895
	6 months	0.29 ± 0.02	0.487	0.36 ± 0.04	0.708	0.705
	12 months	0.36 ± 0.03	0.250	0.37 ± 0.04	0.120	0.135
	24 months	0.37 ± 0.03	0.797	0.39 ± 0.05	0.804	0.843
	36 months	0.37 ± 0.04	0.204	0.41 ± 0.05	0.653	0.234

IOP, intraocular pressure; BCVA, best-corrected visual acuity; TE, trabeculectomy; DS, deep sclerectomy; * Wilcoxon test; ** Mann–Whitney *U*-test.

**Table 3 diagnostics-14-00101-t003:** Development of mean defect of SAP and mean peripapillary RNFL thickness during 3 years of postsurgical follow-up after TE or DS.

		TE	*p* = *	DS	*p* = *	*p* = **
SAP mean defect, dB	Baseline	−11.5 ± 0.9	n.a.	−10.5 ± 0.9	n.a.	0.377
	3 months	−12.3 ± 1.1	0.171	−12.6 ± 1.4	0.068	0.884
	6 months	−11.5 ± 1.0	0.255	−11.1 ± 0.9	0.112	0.922
	12 months	−12.0 ± 1.1	0.090	−11.0 ± 1.0	0.302	0.915
RNFL thickness, μm	Baseline	64.4 ± 2.1	n.a.	64.9 ± 1.9	n.a.	0.963
	3 months	64.5 ± 3.3	0.163	64.6 ± 4.3	0.637	0.819
	6 months	59.8 ± 3.5	0.003	58.7 ± 2.7	0.052	0.829
	12 months	57.2 ± 2.2	<0.001	57.5 ± 3.0	0.016	0.620
	24 months	57.7 ± 2.6	<0.001	58.6 ± 2.5	0.005	0.408
	36 months	59.7 ± 3.5	<0.001	58.4 ± 2.1	<0.001	0.841

SAP, standard automated perimetry; RNFL, retinal nerve fiber layer; TE, trabeculectomy; DS, deep sclerectomy; * Wilcoxon test; ** Mann–Whitney *U*-test.

**Table 4 diagnostics-14-00101-t004:** Number of complications, performed necessary interventions, and secondary glaucoma interventions during 3 years of follow-up after TE or DS.

	TE	DS	*p* = *
Hyphema	4	0	0.008
Corneal erosion	6	2	0.023
Choroidal detachment	16	8	0.001
Bleb leakage	6	2	0.023
Flat anterior chamber	5	3	0.124
Aqueous misdirection syndrome	2	0	0.062
Hypotonia maculopathy	3	3	0.494
Suture lysis (total)	18	0	<0.001
Suture lysis (number of eyes)	15	0	<0.001
Goniopuncture	0	31	<0.001
5-FU (total)	102	67	<0.001
5-FU (number of eyes)	32	15	0.002
Bleb needlings (total)	14	15	0.162
Bleb needlings (number of eyes)	12	15	0.379
Anterior chamber reformation	6	2	0.149
Secondary glaucoma surgery	14	23	0.875

TE, trabeculectomy; DS, deep sclerectomy; * chi-square test.

**Table 5 diagnostics-14-00101-t005:** Percentage of complete and qualified success for resulting IOP levels < 21 mmHg, <18 mmHg, and <15 mmHg in the TE and DS groups after 3 years of follow-up.

		TE	DS	*p* = *
<21 mmHg	Complete success	72%	46%	<0.001
	Qualified success	88%	75%	0.009
<18 mmHg	Complete success	72%	45%	<0.001
	Qualified success	86%	72%	0.006
<15 mmHg	Complete success	70%	40%	<0.001
	Qualified success	80%	62%	0.002

TE, trabeculectomy; DS, deep sclerectomy; * chi-square test.

## Data Availability

The data supporting the reported results can be provided by the corresponding author on reasonable request.

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
