# Peer review of "Functional and Morphological Outcomes after Trabeculectomy and Deep Sclerectomy—Results from a Monocentric Registry Study"

_diagnostics, 2024, doi:10.3390/diagnostics14010101_

Round 1

Reviewer 1 Report

Comments and Suggestions for Authors

I read the manuscript by Pfeiffer et al. with great interest. The study is captivating to read, provides sufficient information regarding the methods used and provides comprehensible tables and figures. Although comparisons of both surgical techniques have been done before, the study adds some new aspects.

I have the three following remarks:

1.) The study provides a variety of functional postoperative data, such as IOP, BCVA, visual field outcomes, complications, and not only OCT data. Therefore, the title in its present form may be a bit misleading. What about „Functional and Morphological Outcomes after Trabeculectomy and Deep Sclerectomy – Results from a Monocentric Registry Study“

2.) Results. Medical files of all eyes undergoing TE or DS between 2014 and 2019 at the University Eye Hospital Bern were reviewed. By how many surgeons were the interventions performed? Were both TE and DS conducted by one (the same?) surgeon or different surgeons? Since the outcome may be surgeon-dependent, this information would be useful.

3.) I suggest to use larger font size for definition of abbreviations below the tables. Likewise, font size should be larger in figure 6.

Comments on the Quality of English Language

Just minor spell check required.

Author Response

Dear editor, dear reviewers,

Thank you very much for reviewing our manuscript “OCT changes after Trabeculectomy and Deep Sclerectomy – Results from a Monocentric Registry Study” and for the helpful advices given. We appreciate your detailed feedback and suggestions. Regarding the points raised in your reviews, we have carefully considered your comments and made the following changes accordingly:

Reviewer #1:

I have the three following remarks:

1.) The study provides a variety of functional postoperative data, such as IOP, BCVA, visual field outcomes, complications, and not only OCT data. Therefore, the title in its present form may be a bit misleading. What about „Functional and Morphological Outcomes after Trabeculectomy and Deep Sclerectomy – Results from a Monocentric Registry Study“

We thank reviewer #1 for this encouraging advice concerning the title of our manuscript. We are of the same opinion as reviewer #1 and changed the title of the manuscript accordingly.

New Title: “Functional and Morphological Outcomes after Trabeculectomy and Deep Sclerectomy – Results from a Monocentric Registry Study”.

2.) Results. Medical files of all eyes undergoing TE or DS between 2014 and 2019 at the University Eye Hospital Bern were reviewed. By how many surgeons were the interventions performed? Were both TE and DS conducted by one (the same?) surgeon or different surgeons? Since the outcome may be surgeon-dependent, this information would be useful.

We thank reviewer #1 for this valuable piece of advice. The surgical procedures were performed by four different ophthalmic surgeons. All four surgeons were experienced and regularly performed both procedures described in the manuscript (i.e. Trabeculectomy and Deep Sclerectomy). For reasons of clarity we added this information to the manuscript in the following manner in the materials and methods section:

All surgical procedures were performed by four experienced ophthalmic surgeons. Both, TE and DS were regularly performed by each of the four surgeons. 

3.) I suggest to use larger font size for definition of abbreviations below the tables. Likewise, font size should be larger in figure 6.

Indeed, font size in the definitions under the tables and in Figure 6 are hardly readable. We increased font size accordingly in the manuscript.

Reviewer 2 Report

Comments and Suggestions for Authors

This study aimed to compare the efficacy of trabeculectomy and deep sclerectomy in lowering intraocular pressure and thereby preserving visual field and peripapillary retinal nerve fiber layer in primary open angle glaucoma participants. The study found that both techniques demonstrated comparable efficacy regarding postoperative reduction of intraocular pressure and medication. I have just a few comments and suggestions for improvement as outlined below:

1)    In the introduction, although the authors have briefly touched on comparison of potential surgical complications of each of the techniques, it may be beneficial to expand on this further and perhaps include data on the chances of these complications occurring.

2)    When reporting the means of values with associated spread, can the authors please specify what the +/- values represent? Is it standard error, standard deviation, etc.

3)    Although SAP average mean deviation was reported, it would benefit readers to include an extra figure illustrating the distribution of all mean deviation values in the sample at baseline, to give readers an idea of what this distribution looks like. Perhaps something like a histogram to represent this would help.

4)    Were there any reliability criteria cutoffs for visual field testing?

Author Response

Dear editor, dear reviewers,

Thank you very much for reviewing our manuscript “OCT changes after Trabeculectomy and Deep Sclerectomy – Results from a Monocentric Registry Study” and for the helpful advices given. We appreciate your detailed feedback and suggestions. Regarding the points raised in your reviews, we have carefully considered your comments and made the following changes accordingly:

Reviewer #2:

I have just a few comments and suggestions for improvement as outlined below:

1)    In the introduction, although the authors have briefly touched on comparison of potential surgical complications of each of the techniques, it may be beneficial to expand on this further and perhaps include data on the chances of these complications occurring.

The most frequent complications occurring after TS and DS are hypotony, choroidal effusion, cataract development and a shallow anterior chamber. All these complications may lead to (long-term) loss of visual function and can make further surgical and non-surgical interventions necessary. In a systemic review comparing TE and DS published by Rulli et al described the already published data from a number of different studies and compared them concerning these complications. They found that the risk for all these complications was higher in the TE group. The comparisons performed for this review are quite extensive. We added the following sentences to the introduction section but omitted reporting the exact calculated relative risks but gave the reference. We hope reviewer #2 is content with this:

Rulli et al. performed a systematic review of a large number of already published studies and compared the reported rates for the most frequently occurring complications after TE and DS[7]. They found that the probability for hypotony, choroidal effusion, flat anterior chamber and cataract development is higher after TE than DS.

2)    When reporting the means of values with associated spread, can the authors please specify what the +/- values represent? Is it standard error, standard deviation, etc.

We thank reviewer #2 for this question. The reported variables were all given as mean +/- standard error of mean (SEM). We further specified this in the materials and methods section in the following way:

For description, continuous variables are given as mean and standard error of mean.

However, when going through the manuscript and the reported data we found that mean patient age was not mean value +/- SEM. We therefore adjusted for this mistake in the new version of the manuscript.

3)    Although SAP average mean deviation was reported, it would benefit readers to include an extra figure illustrating the distribution of all mean deviation values in the sample at baseline, to give readers an idea of what this distribution looks like. Perhaps something like a histogram to represent this would help.

We thank reviewer #2 for this helpful advice to further clarify and describe the group of patients included for this analysis. We analyzed the distribution of SAP results collected at baseline before surgery and created a histogram accordingly. We grouped the SAP results in the following way. MD below <5 dB were grouped as early glaucomatous cases, MD between 5 and 12 dB were grouped together and termed moderate glaucoma cases, MD between 12 and 20 dB were grouped and named advanced disease stages and finally all MD > 20dB were grouped together in the end stage glaucoma group. The histogram has been added to the manuscript as new figure 4 and the numbers of the following figures have been changed accordingly. The changes made in the manuscript are as follows:

Mean deviation measured using standard automated perimetry (SAP) was compa-rable between groups at baseline (TE: -11.5±0.9 dB; DS: -10.5±0.9 dB; p= 0.377 / for MD distribution before surgery see Figure 4).

The added figure is the following:

4)    Were there any reliability criteria cutoffs for visual field testing?         

We thank reviewer #2 for this central question concerning the applied SAP reliability criteria. We added the SAP results to our analysis only when reliability criteria were met by the patient / the performed test. The cut-off values were a false-positive and a false-negative rate of <20%. For clarification we added the following description to the materials and methods section:

Visual field test results were only used for analysis when reliability criteria were met. These were a false-positive and false-negative rate of <20%.

Finally, we have read and corrected minor spelling and language mistakes.